# TransMatcher: Deep Image Matching Through Transformers for Generalizable Person Re-identification

**Shengcai Liao**[*] **and Ling Shao**
Inception Institute of Artificial Intelligence (IIAI), Abu Dhabi, UAE
`https://liaosc.wordpress.com/`

## Abstract

Transformers have recently gained increasing attention in computer vision. However, existing studies mostly use Transformers for feature representation learning, e.g. for image classification and dense predictions, and the generalizability of Transformers is unknown. In this work, we further investigate the possibility of applying Transformers for image matching and metric learning given pairs of images. We find that the Vision Transformer (ViT) and the vanilla Transformer with decoders are not adequate for image matching due to their lack of image-to-image attention. Thus, we further design two naive solutions, i.e. query-gallery concatenation in ViT, and query-gallery cross-attention in the vanilla Transformer. The latter improves the performance, but it is still limited. This implies that the attention mechanism in Transformers is primarily designed for global feature aggregation, which is not naturally suitable for image matching. Accordingly, we propose a new simplified decoder, which drops the full attention implementation with the softmax weighting, keeping only the query-key similarity computation. Additionally, global max pooling and a multilayer perceptron (MLP) head are applied to decode the matching result. This way, the simplified decoder is computationally more efficient, while at the same time more effective for image matching. The proposed method, called TransMatcher, achieves state-of-the-art performance in generalizable person re-identification, with up to 6.1% and 5.7% performance gains in Rank-1 and mAP, respectively, on several popular datasets. Code is available at `https://github.com/ShengcaiLiao/QAConv`.

## 1 Introduction

The Transformer [24] is a neural network based on attention mechanisms. It has shown great success in the field of natural language processing. Recently, it has also shown promising performance for computer vision tasks, including image classification [7, 14], object detection [2, 37, 14, 26], and image segmentation [14, 26], thus gaining increasing attention in this field. However, existing studies mostly use Transformers for feature representation learning, e.g. for image classification or dense predictions, and the generalizability of Transformers is unknown. At a glance, query-key similarities are computed by dot products in the attention mechanisms of Transformers. Therefore, these models could potentially be useful for image matching. In this work, we further investigate the possibility of applying Transformers for image matching and metric learning given pairs of images, with applications in generalizable person re-identification.

Attention mechanisms are used to gather global information from different locations according to query-key similarities. The vanilla Transformer [24] is composed of an encoder that employs

---

[*]Shengcai Liao is the corresponding author.

35th Conference on Neural Information Processing Systems (NeurIPS 2021).

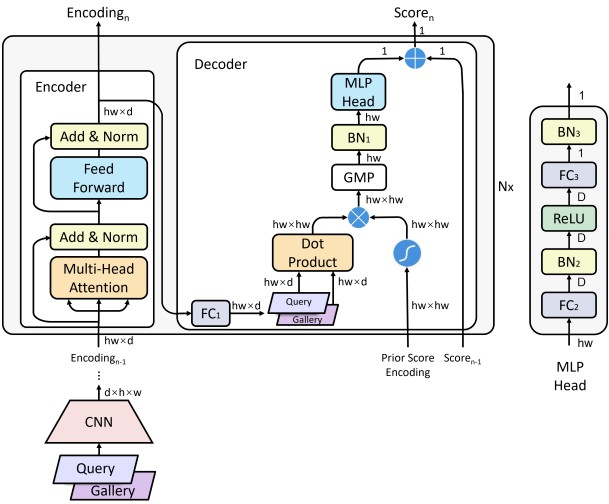

Figure 1: The structure of the proposed TransMatcher for image matching. A standard Transformer encoder without positional encoding is used for feature encoding. Then, query and gallery encodings are matched by a dot product. Global max pooling (GMP) is applied to find the optimal matching scores and locations, and an MLP head is appended to produce the final matching scores. Note that the batch dimension is ignored in this figure for simplicity.

self-attention, and a decoder that further incorporates a cross-attention module. The difference is that the query and key are the same in the self-attention, while they are different in the cross-attention. The Vision Transformer (ViT) [7] applies a pure Transformer encoder for feature learning and image classification. While the Transformer encoder facilitates feature interaction among different locations of the same image, it cannot address the image matching problem being studied in this paper, because it does not enable interaction between different images. In the decoder, however, the cross-attention module does have the ability for cross interaction between query and the encoded memory. For example, in the decoder of the detection Transformer (DETR) [2], learnable query embeddings are designed to decode useful information in the encoded image memory for object localization. However, the query embeddings are independent from the image inputs, and so there is still no interaction between pairs of input images. Motivated by this, how about using actual image queries instead of learnable query embeddings as input to decoders?

Person re-identification is a typical image matching and metric learning problem. In a recent study called QAConv [10], it was shown that explicitly performing image matching between pairs of deep feature maps helps the generalization of the learned model. This inspires us to investigate the capability and generalizability of Transformers for image matching and metric learning between pairs of images. Since training through classification is also a popular strategy for metric learning, we start from a direct application of ViT and the vanilla Transformer with a powerful ResNet [3] backbone for person re-identification. However, this results in poor generalization to different datasets. Then, we consider formulating explicit interactions between query[2] and gallery images in Transformers. Two naive solutions are thus designed. The first one uses a pure Transformer encoder, as in ViT, but concatenates the query and gallery features together as inputs, so as to enable the self-attention module to read both query and gallery content and apply the attention between them. The second design employs the vanilla Transformer, but replaces the learnable query embedding in the decoder by the ready-to-use query feature maps. This way, the query input acts as a real query from the actual retrieval inputs, rather than a learnable query which is more like a prior or a template. Accordingly, the cross-attention module in the decoder is able to gather information across query-key pairs, where the key comes from the encoded memory of gallery images.

While the first solution does not lead to improvement, the second one is successful with notable performance gain. However, compared to the state of the art in generalizable person re-identification,

---

[2]Query/gallery in person re-identification and query/key or target/memory in Transformers have very similar concepts originated from information retrieval. We use the same word query here in different contexts.

the performance of the second variant is still not satisfactory. We further consider that the attention mechanism in Transformers might be primarily for global feature aggregation, which is not naturally suitable for image matching, though the two naive solutions already enable feature interactions between query and gallery images. Therefore, to improve the effectiveness of image matching, we propose a new simplified decoder, which drops the full attention implementation with the softmax weighting, keeping only the query-key similarity computation. Additionally, inspired from QAConv [10], global max pooling (GMP) is applied, which acts as a hard attention to gather similarity values, instead of a soft attention to weight feature values. This is because, in image matching, we are more interested in matching scores than feature values. Finally, a multilayer perceptron (MLP) head maps the matching result to a similarity score for each query-gallery pair. This way, the simplified decoder is computationally more efficient, while at the same time more effective for image matching.

We call the above design TransMatcher (see Fig. 1), which targets at efficient image matching and metric learning in particular. The contributions of this paper are summarized as follows.

- We investigate the possibility and generalizability of applying Transformers for image matching and metric learning, including direct applications of ViT and the vanilla Transformer, and two solutions adapted specifically for matching images through attention. This furthers our understanding of the capability and limitation of Transformers for image matching.

- According to the above, a new simplified decoder is proposed for efficient image matching, with a focus on similarity computation and mapping.

- With generalizable person re-identification experiments, the proposed TransMatcher is shown to achieve state-of-the-art performance on several popular datasets, with up to 6.1% and 5.7% performance gains in Rank-1 and mAP, respectively.

## 2 Related Work

Given pairs of input images, deep feature matching has been shown to be effective for person re-identification. Li et al. [8] proposed a novel filter pairing neural network (FPNN) to handle misalignment and occlusions in person re-identification. Ahmed et al. [1] proposed a local neighborhood matching layer to match deep feature maps of query and gallery images. Suh et al. [21] proposed a deep neural network to learn part-aligned bilinear representations for person re-identification. Shen et al. [18] proposed a Kronecker-product matching (KPM) module for matching person images in a softly aligned way. Liao and Shao [10] proposed the query adaptive convolution (QAConv) for explicit deep feature matching, which is proved to be effective for generalizable person re-identification. They further proposed a graph sampler (GS) for efficient deep metric learning [11].

Generalizable person re-identification has gained increasing attention in recent years. Zhou et al. [36] proposed the OSNet, and showed that this new backbone network has advantages in generalization. Jia et al. [5] applied IBN-Net-b [15] together with a feature normalization to alleviate both style and content variance across datasets to improve generalizability. Song et al. [20] proposed a domain-invariant mapping network (DIMN) and further introduced a meta-learning pipeline for effective training and generalization. Qian et al. [17] proposed a deep architecture with leader-based multi-scale attention (MuDeep), with improved generalization of the learned models. Yuan et al. [31] proposed an adversarial domain-invariant feature learning network (ADIN) to separate identity-related features from challenging variations. Jin et al.[6] proposed a style normalization and restitution module, which shows good generalizability for person re-identification. Zhuang et al. [38] proposed a camera-based batch normalization (CBN) method for domain-invariant representation learning, which utilizes unlabeled target data to adapt the BN layer in a quick and unsupervised way. Wang et al. [27] created a large-scale synthetic person dataset called RandPerson, and showed that models learned from synthesized data generalize well to real-world datasets. However, current methods are still far from satisfactory in generalization for practical person re-identification.

There are a number of attentional networks [12, 16, 13, 30, 19, 9, 29, 32, 4] proposed for person re-identification, but focusing on representation learning. More recently, Zhao et al. [33] proposed a cross-attention network for person re-identificaiton. However, it is still applied for feature refinement, instead of explicit image matching between gallery and probe images studied in this paper.

Transformers have recently received increasing attention for computer vision tasks, including image classification [7, 14], object detection [2, 37, 14, 26], image segmentation [14, 26], and so on. For

example, ViT was proposed in [7], showing that a pure Transformer-based architecture is capable of effective image classification. DETR was proposed in [2], providing a successful end-to-end Transformer solution for object detection. Later, several studies, such as the Deformable DETR [37], Swin [14], and PVT [26], improved the computation of Visual Transformers and further boosted their performance. However, existing studies mostly use Transformers for feature representation learning, e.g. for image classification and dense predictions. There lacks a comprehensive study on whether Transformers are effective for image matching and metric learning and how its capability is in generalizing to unknown domains.

## 3  Transformers

For the vanilla Transformer [24], the core module is the multi-head attention (MHA). First, a scaled dot-product attention is defined as follows:

$$\text{Attention}(Q, K, V) = \text{softmax}(\frac{QK^T}{\sqrt{d_k}})V, \tag{1}$$

where $Q \in \mathbb{R}^{T \times d_k}$ is the query (or target) matrix, $K \in \mathbb{R}^{M \times d_k}$ is the key (or memory) matrix, $V \in \mathbb{R}^{M \times d_v}$ is the value matrix, $T$ and $M$ are the sequence lengths of the query and key, respectively, $d_k$ is the feature dimension of the query and key, and $d_v$ is the feature dimension of $V$. In visual tasks, $Q$ and $K$ are usually reshaped query and key feature maps, with $T = M = hw$, where $h$ and $w$ are the height and width of the query and key feature maps, respectively. Then, the MHA is defined as:

$$\text{head}_i = \text{Attention}(QW_i^Q, KW_i^K, VW_i^V), \tag{2}$$

$$\text{MultiHead}(Q, K, V) = \text{Concat}(\text{head}_1, \ldots, \text{head}_H)W^O, \tag{3}$$

where $W_i^Q \in \mathbb{R}^{d \times d_k}$, $W_i^K \in \mathbb{R}^{d \times d_k}$, $W_i^V \in \mathbb{R}^{d \times d_v}$, and $W^O \in \mathbb{R}^{hd_v \times d}$ are parameter matrices, and $H$ is the number of heads. Then, $Q = K = V$ in the multi-head self-attention (MHSA) in the encoders, while they are defined separately in the multi-head cross-attention (MHCA) in the decoders.

The structure of the Transformer encoder without positional encoding is shown on the left of Fig. 1. Beyond MHSA, it further appends a feed-forward layer to first increase the feature dimension from $d$ to $D$, and then recover it back from $D$ to $d$. Besides, the encoder can be self-stacked $N$ times, where $N$ is the total number of encoder layers. In ViT [7], only Transformer encoders are used, and positional encoding is further applied. In the vanilla Transformer [24], decoders with MHCA are further applied, with the query being learnable query embeddings initially, and the output of the previous decoder layer later on, and the key and value being the output of the encoder layer. The decoder can also be self-stacked $N$ times.

## 4  Image Matching with Transformers: Naive Solutions

While the above ViT and vanilla Transformer are able to perform image matching through black-box feature extraction and distance learning, they are not optimal for this task because they lack image-to-image interaction in their designs. Though cross-attention is employed in the Transformer decoders, in its original form the query either comes from learnable query embeddings, or from the output of the previous decoder layer.

Therefore, we adapt Transformers with two naive solutions for image matching and metric learning. Building upon a powerful ResNet [3] backbone, the first solution appends ViT, but not simply for feature extraction. Instead, a pair of query and gallery feature maps are concatenated to double the sequence length, forming a single sample for the input of ViT. Thus, both the query and key for the self-attention layer contain query image information in one half and gallery image information in the other half. Therefore, the attention computation in Eq. (1) is able to interact query and gallery inputs for image matching. This variant is denoted as Transformer-Cat.

The second solution appends the vanilla Transformer, but instead of learnable query embeddings, ResNet query features are directly input into the first decoder. This way, the cross-attention layer in the decoders is able to interact the query and gallery samples being matched. This variant is denoted as Transformer-Cross.

The structure of these two variants can be found in the Appendix. Note that these two solutions have high computational and memory costs, especially for large $d$, $D$, and $N$ (*c.f.* Section 6.4).

# 5 The Proposed TransMatcher

Though the above two solutions enable query-gallery interaction in the attention mechanism for image matching, they are not adequate for distance metric learning. This is because, taking a deeper look at Eq. (1) for the attention, it can be observed that, though similarity values between $Q$ and $K$ are computed, they are only used for softmax-based weighting to aggregate features from $V$. Therefore, the output of the attention is always a weighted version of $V$ (or $K$), and thus cross-matching between a pair of inputs is not directly formulated.

To address this, we propose a simplified decoder, which is explicitly formulated towards similarity computation. The structure of this decoder is shown in the middle of Fig. 1. First, both gallery and query images are independently encoded by $N$ sequential Transformer encoders after a backbone network, as shown on the left of Fig. 1. This encoding helps aggregating global information from similar body parts for the subsequent matching step. The resulting feature encodings are denoted by $Q_n \in \mathbb{R}^{hw \times d}$ and $K_n \in \mathbb{R}^{hw \times d}$, $n = 1, \ldots, N$, for the query and gallery, respectively. Then, as in Eq. (2), both the gallery and query encodings are transformed by a fully connected (FC) layer $FC_1$:

$$Q'_n = Q_n W_n, K'_n = K_n W_n, \tag{4}$$

where $W_n \in \mathbb{R}^{d \times d}$ is the parameter matrix for encoder-decoder layer $n$. Different from Eq. (2), we use shared FC parameters for both query and gallery, because they are exchangeable in the image matching task, and the similarity metric needs to be symmetrically defined. Then, the dot product is computed between the transformed features, as in Eq. (1):

$$S_n = Q'_n K'_n{}^T, \tag{5}$$

where $S_n \in \mathbb{R}^{hw \times hw}$ are the similarity scores. In addition, a learnable prior score embedding $R \in \mathbb{R}^{hw \times hw}$ is designed, which defines prior matching scores between different locations of query and gallery images. Then, it is used to weight the similarity values:

$$S'_n = S_n * \sigma(R), \tag{6}$$

where $*$ denotes element-wise multiplication, and $\sigma$ is the sigmoid function to map the prior score embedding into weights in $[0, 1]$.

After that, a GMP layer is applied along the last dimension of $hw$ elements:

$$S''_n = \max(S'_n, \text{dim=-1}). \tag{7}$$

This way, the optimal local matching over all key locations is obtained, as in QAConv [10]. Compared to Eq. (1), the GMP here can be considered as a hard attention, but it is used for similarity matching rather than softmax-based feature weighting like in the soft attention. Note that multi-head design in MHA is not considered here (*c.f.* Section 6.6).

Then, after a batch normalization layer $BN_1$, an MLP head is further appended, similar to the feed-forward layer of Transformers. It is composed of $MLPHead_1 = (FC_2, BN_2, ReLU)$ to map the $hw$ similarity values to dimension $D$, and $MLPHead_2 = (FC_3, BN_3)$ to map dimension $D$ to 1 as a single output score $S'''_n$.

Finally, decoder $n$ outputs a similarity score by fusing the output of the previous decoder:

$$S''''_n = S'''_n + S''''_{n-1}, \tag{8}$$

where $S''''_0$ is defined as 0. With $N$ stacked encoder-decoder blocks, as shown in Fig. 1, this can be considered as residual similarity learning. Note that the stack of encoder-decoder blocks in TransMatcher is different from that in the vanilla Transformer. In TransMatcher, the encoder and decoder are connected before being stacked, while in the vanilla Transformer they are stacked independently before connection. This way, the decoder of TransMatcher is able to perform cross matching with different levels of encoded features for residual similarity learning.

However, the GMP operation in Eq. (7) is not symmetric. To make TransMatcher symmetric for the query and gallery, the GMP operation in Eq. (7) can also be applied along dim=0; that is, conduct an inverse search of best matches over all query locations. Keeping other operations the same, this will result in another set of similarity scores, which are summed with the original ones after the $FC_3$ layer. Further details can be found in the Appendix. Note that this is not reflected in Fig. 1 for simplicity of illustration.

Finally, the outputs of TransMatcher scores for all query-gallery pairs in a batch are collected for pairwise metric learning following the same pipeline in QAConv-GS [11], and the same binary cross entropy loss is used as in the QAConv-GS.

# 6 Experiments

## 6.1 Datasets

Four large-scale person re-identification datasets, CUHK03 [8], Market-1501 [34], MSMT17 [28], and RandPerson [27], which are publicly available for research purpose, are used in our experiments. The CUHK03 dataset includes 1,360 persons and 13,164 images,with 767 and 700 subjects used for training and testing, respectively, as in the CUHK03-NP protocol [35]. Besides, the "detected" subset is used, which is more challenging than the "labeled" subset. The Market-1501 dataset contains 32,668 images of 1,501 identities captured from six cameras, with 12,936 images from 751 identities for training, and 19,732 images from 750 identities for testing.MSMT17 includes 4,101 identities and 126,441 images captured from 15 cameras, with 32,621 images from 1,041 identities for training, and the remaining images from 3,010 identities for testing. RandPerson is a recently released synthetic person re-identification dataset for large-scale training towards generalization testing. It is with 8,000 persons and 1,801,816 images. A subset with 132,145 images of the 8,000 IDs is used for training.

Cross-dataset evaluation is performed on these datasets by training on the training subset of one dataset, and evaluating on the test subsets of other datasets. Except that for MSMT17 we further use an additional setting with all images for training, regardless of the subset splits. This is denoted by $\text{MSMT17}_{all}$. All evaluations follow the single-query evaluation protocol. The Rank-1 (Top1) accuracy and mean average precision (mAP) are used as the performance evaluation metrics.

## 6.2 Implementation Details

The implementation of TransMatcher is built upon the official PyTorch project of QAConv-GS [3] [11], as the graph sampler (GS) proposed in this project is efficient for metric learning and quite suitable for the learning of TransMatcher. We keep most of the settings the same as QAConv-GS. Specifically, ResNet-50 [3] is used as the backbone network, with three instance normalization (IN) [23] layers further appended as in IBN-Net-b [15], following several recent studies [5, 36, 6, 38, 11]. The backbone network is pre-trained on ImageNet, with the states of the BN layers being fixed. The layer3 feature map is used, with a $3\times3$ neck convolution appended to produce the final feature map. The input image is resized to $384 \times 128$. The batch size is set to 64, with K=4 for the GS sampler. The network is trained with the SGD optimizer, with a learning rate of 0.0005 for the backbone network, and 0.005 for newly added layers. They are decayed by 0.1 after 10 epochs, and 15 epochs are trained in total. Except that for RandPerson [27] the total number of epochs is 4, and the learning rate step size is 2, according to the experiences in [27, 11]. Gradient clipping is applied with $T = 4$ [11]. Several commonly used data augmentation methods are applied, including random flipping, cropping, occlusion, and color jittering. All experiments are run on a single NVIDIA V100 GPU.

For the proposed TransMatcher, unless otherwise indicated, $d$=512 and $D$=2048 by default as in the original Transformer [24], and $H$=1 and $N$=3 for higher efficiency. Please refer to Section 6.6 for further parameter analysis. Besides, in practice, we find that when $N$ decoders are used, using $N - 1$ encoders together with the ResNet feature map directly pairing the first decoder slightly improves the results while being more efficient, which is preferred in the implementation (*c.f.* Appendix).

## 6.3 Comparison to the State of the Art

A comparison to the state of the art (SOTA) in generalizable person re-identification is shown in Table 1. Several methods published very recently for generalizable person re-identification are compared, including OSNet [36], MuDeep [17], ADIN [31], SNR [6], CBN [38], QAConv [10], and QAConv-GS [11]. From Table 1 it can be observed that TransMatcher significantly improves the previous SOTA. For example, with Market-1501 for training, the Rank-1 and mAP are improved by 5.8% and 5.7% on CUHK03-NP, respectively, and they are improved by 6.1% and 3.4% on MSMT17, respectively. With MSMT17 → Market-1501, the improvements are 5.0% for Rank-1 and 5.3% for mAP. With the synthetic dataset RandPerson for training, the improvements on Market-1501 are 3.3% for Rank-1 and 5.3% for mAP, and the gains on MSMT17 are 5.9% for Rank-1 and 3.3% for mAP.

Compared to the second best method QAConv-GS, since it shares the same code base and training setting with the proposed TransMatcher, it indicates that TransMatcher is a superior image matching

---

[3]QAConv-GS project under MIT License: `https://github.com/ShengcaiLiao/QAConv`.

Table 1: Comparison of the state-of-the-art direct cross-dataset evaluation results (%). $MSMT_{all}$ means all images are used for training, regardless of the subset splits.

| Method | Venue | Train Set | CUHK03-NP | | Market-1501 | | MSMT17 | |
|---|---|---|---|---|---|---|---|---|
| | | | Rank-1 | mAP | Rank-1 | mAP | Rank-1 | mAP |
| MGN [25, 17] | MM'18 | Market | 8.5 | 7.4 | - | - | - | - |
| MuDeep [17] | PAMI'20 | Market | 10.3 | 9.1 | - | - | - | - |
| CBN [38] | ECCV'20 | Market | - | - | - | - | 25.3 | 9.5 |
| QAConv [10] | ECCV'20 | Market | 9.9 | 8.6 | - | - | 22.6 | 7.0 |
| QAConv-GS [11] | arXiv'21 | Market | 16.4 | 15.7 | - | - | 41.2 | 15.0 |
| TransMatcher | Ours | Market | **22.2** | **21.4** | - | - | **47.3** | **18.4** |
| PCB [22, 31] | ECCV'18 | MSMT | - | - | 52.7 | 26.7 | - | - |
| MGN [25, 31] | MM'18 | MSMT | - | - | 48.7 | 25.1 | - | - |
| ADIN [31] | WACV'20 | MSMT | - | - | 59.1 | 30.3 | - | - |
| SNR [6] | CVPR'20 | MSMT | - | - | 70.1 | 41.4 | - | - |
| CBN [38] | ECCV'20 | MSMT | - | - | 73.7 | 45.0 | - | - |
| QAConv-GS [11] | arXiv'21 | MSMT | 20.0 | 19.2 | 75.1 | 46.7 | - | - |
| TransMatcher | Ours | MSMT | **23.7** | **22.5** | **80.1** | **52.0** | - | - |
| OSNet [36] | CVPR'19 | $MSMT_{all}$ | - | - | 66.5 | 37.2 | - | - |
| QAConv [10] | ECCV'20 | $MSMT_{all}$ | 25.3 | 22.6 | 72.6 | 43.1 | - | - |
| QAConv-GS [11] | arXiv'21 | $MSMT_{all}$ | 27.2 | 27.1 | 80.6 | 55.6 | - | - |
| TransMatcher | Ours | $MSMT_{all}$ | **31.9** | **30.7** | **82.6** | **58.4** | - | - |
| RP Baseline [27] | MM'20 | RandPerson | 13.4 | 10.8 | 55.6 | 28.8 | 20.1 | 6.3 |
| QAConv-GS [11] | arXiv'21 | RandPerson | 14.8 | 13.4 | 74.0 | 43.8 | 42.4 | 14.4 |
| TransMatcher | Ours | RandPerson | **17.1** | **16.0** | **77.3** | **49.1** | **48.3** | **17.7** |

Table 2: Comparison of different Transformers trained on MSMT17 for direct cross-dataset evaluation (%). mAcc (%) is the average of all Rank-1 and mAP results on both CUHK03-NP and Market-1501 over four random runs.

| Method | $d$ | $D$ | N | Time (h) | CUHK03-NP | | Market-1501 | | mAcc |
|---|---|---|---|---|---|---|---|---|---|
| | | | | | Rank-1 | mAP | Rank-1 | mAP | |
| ViT | 512 | 2048 | 3 | **0.99** | 12.0 | 12.4 | 57.7 | 29.8 | 27.42 |
| Transformer | 512 | 2048 | 3 | 1.16 | 13.2 | 13.3 | 54.3 | 29.0 | 27.01 |
| TransMatcher | 512 | 2048 | 3 | 1.44 | **23.7** | **22.5** | **80.1** | **52.0** | **44.29** |
| Transformer-Cat | 128 | 512 | 2 | 4.89 | 13.1 | 13.2 | 53.9 | 27.4 | 25.34 |
| Transformer-Cross | 128 | 512 | 2 | 3.48 | 18.9 | 19.8 | 66.2 | 40.1 | 36.70 |
| TransMatcher | 128 | 512 | 2 | **1.11** | **22.5** | **21.4** | **77.4** | **49.3** | **42.12** |

and metric learning method for generalizable person re-identification, thanks to the effective cross-matching design in the new decoders.

## 6.4 Comparison of Transformers

A comparison of different Transformers trained on MSMT17 for direct cross-dataset evaluation is shown in Table 2. For a fair comparison, they are all trained with the same settings as described in Section 6.2. Besides, $H$=1 for all models. ViT, the vanilla Transformer, and TransMatcher all have the same parameter settings. Though we use an NVIDIA V100 GPU with 32GB of memory, Transformer-Cat and Transformer-Cross still encounter the memory overflow problem under the same parameter settings as TransMatcher. Therefore, we have to set $d$=128, $D$=512, and $N$=2 for them to run, and accordingly, a smaller version of TransMatcher with the same set of parameters is also provided for comparison.

From the results shown in Table 2, it can be observed that ViT and the vanilla Transformer perform poor in generalizing to other datasets. In contrast, the proposed TransMatcher significantly improves the performance. This confirms that simply applying Transformers for the image matching task is not effective, because they lack cross-image interaction in their designs.

Table 3: Ablation of different components in TransMatcher. Training is performed on MSMT17. mAcc (%) is the average of all Rank-1 and mAP results on all test sets over four random runs. PriorEmbed is the prior score embedding, while PosEmbed is the positional embedding.

| $FC_1$ | $BN_1$ | $MLPHead_1$ | $MLPHead_2$ | PriorEmbed | PosEmbed | mAcc |
|---|---|---|---|---|---|---|
| | | | ✓ | | | 41.44 |
| | | ✓ | ✓ | | | 42.82 |
| | ✓ | ✓ | ✓ | | | 43.66 |
| ✓ | | ✓ | ✓ | | | 43.70 |
| ✓ | ✓ | ✓ | ✓ | | | 44.04 |
| ✓ | ✓ | ✓ | ✓ | ✓ | | **44.29** |
| ✓ | ✓ | ✓ | ✓ | | ✓ | 42.39 |
| ✓ | ✓ | ✓ | ✓ | ✓ | ✓ | 42.56 |

Besides, we find that Transformer-Cat does not lead to improvement compared to ViT and the vanilla Transformer. It is a smaller model, though. However, Transformer-Cross does lead to notable improvements, indicating that the cross-matching of gallery and query images in Transformer decoders is potentially more effective. However, it is still not as good as the smaller version of TransMatcher. For example, on Market-1501, TransMatcher improves the Rank-1 by 11.2% and the mAP by 9.2% over the Transformer-Cross. Therefore, the cross-attention design in the original Transformers is not efficient enough for image matching, due to its focus on feature aggregation but not similarity matching. More variants and experiments of Transformers can be found in Appendix.

As for the running speed, the training times of these methods are also listed in Table 2. As can be seen, without cross-matching, ViT is the most efficient, followed by the vanilla Transformer. TransMatcher is not as efficient as ViT due to the explicit cross-matching between query and gallery images. However, it is still acceptable, thanks to the new simplified decoder. In contrast, even with a small set of parameters, Transformer-Cat and Transformer-Cross are still quite heavy to compute.

### 6.5 Ablation Study

The structure of the proposed TransMatcher shown in Fig. 1 is carefully ablated, with results listed in Table 3. The training is performed on MSMT17. For ease and reliable comparison, we report the average of all Rank-1 and mAP results on all test sets over four random runs. This is denoted by mAcc. We start with Dot Product + GMP + $MLPHead_2$ (the input dimension to $FC_3$ needs to be adapted to $hw$ accordingly), which is the simplest and most necessary configuration. Then, by adding $MLPHead_1$, the performance is improved by 1.38%, indicating that increasing the dimension to $D$, as in Transformers, is useful. Then, by including $FC_1$ / $BN_1$ independently, the performance gain is 0.84% / 0.88%, and by including them together, the performance can be further improved. Finally, when the prior score embedding is appended, the best performance is achieved. Interestingly, when we include a learnable positional embedding in the encoders, as in ViT, either independently or together with the prior score embedding, the performance is degraded. This indicates that mixing the position information with visual features for image matching is not useful in our design. In contrast, learning spatial-aware prior matching scores separately for score weighting is more effective. More ablation study and analysis can be found in the Appendix.

### 6.6 Parameter Analysis

To understand the parameter selection of the proposed TransMatcher, we train it on MSMT17 with different parameter configurations to the defaults, with the mAcc results as well as the training time shown in Fig. 2. First, the performance is gradually improved by increasing the model dimension $d$. However, the training time is also increased quadratically. Therefore, to provide a balance between accuracy and running speed, $d$=512 is selected, which is the same as in the vanilla Transformer [24].

For the feed forward-dimension $D$, the performance is also gradually improved when increasing the value. However, the training time is less affected, because the feed-forward operation is only applied after the dot product and GMP, where the dimension of $d$ and one spatial dimension $hw$ are already contracted. Nevertheless, large $D$ will increase the memory usage. Therefore, $D$=2048 is selected, which is also the same as in the vanilla Transformer [24].

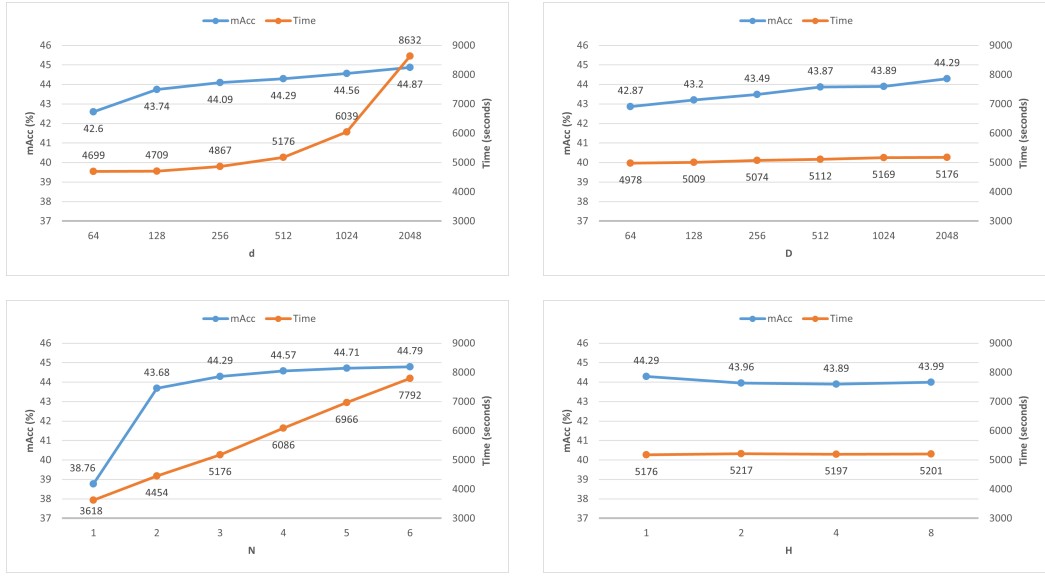

Figure 2: Parameter analysis of the proposed TransMatcher.

As for the number of layers $N$, the performance is also gradually improved with increasing $N$. However, after $N$=3 the performance tends to saturate, and the training time grows linearly with the increasing number of layers. Therefore, $N$=3 is a reasonable balance for our choice. In addition, with $N = 1$ there is no encoder used (for details please see Appendix), and from Fig. 2 it is clear that this is inferior, indicating that including an encoder is important. On the other hand, from the poor performance of ViT where there are only encoders, it is clear that the decoder is also important.

Finally, for the number of heads $H$ in the encoders, it appears that larger $H$ does not lead to improved results. Since the training time is also not affected, we simply select $H$=1 in the encoders, and do not implement the multi-head mechanism in the decoders.

### 6.7 Qualitative Analysis

With the help of the GMP layer, inspired from QAConv [10], the proposed TransMatcher is able to find the best local correspondence matches in each decoder layer. Some qualitative matching results are shown in Fig. 3 for a better understanding of TransMatcher. More examples can be found in the Appendix. The model used here is trained on the MSMT17 dataset [28], and the evaluations are done on the query subset of the Market-1501 dataset [34]. Results of both positive pairs and hard negative pairs are shown. For a clear illustration, only reliable correspondences with matching scores over a certain threshold are shown, where the threshold is determined by a false acceptance rate of 1‰ over all matches of negative pairs. Note that the local positions are coarse due to the $24 \times 8$ size of the feature map.

As can be observed from Fig. 3, the proposed method is able to find correct local correspondences for positive pairs of images, even if there are notable misalignments in both scales and positions, pose, viewpoint, and illumination variations, occlusions, and low resolution blur. Besides, for hard negative pairs, the matching of TransMatcher still appears to be mostly reasonable, by linking visually similar parts or even the same person who might be incorrectly labeled.

This indicates that the proposed TransMatcher is effective in local correspondence matching, and note that it learns to do this with the only supervision of identity information. Besides, the matching capability is generalizable to other datasets beyond the training set. From the illustration it can also be seen that, generally, matching results of the first decoder layer are not as successful as the next two layers, and the matching with the last decoder layer appears to be the best. This indicates that both Transformer encoders and decoders helps the model to match better by aggregating global similarity information.

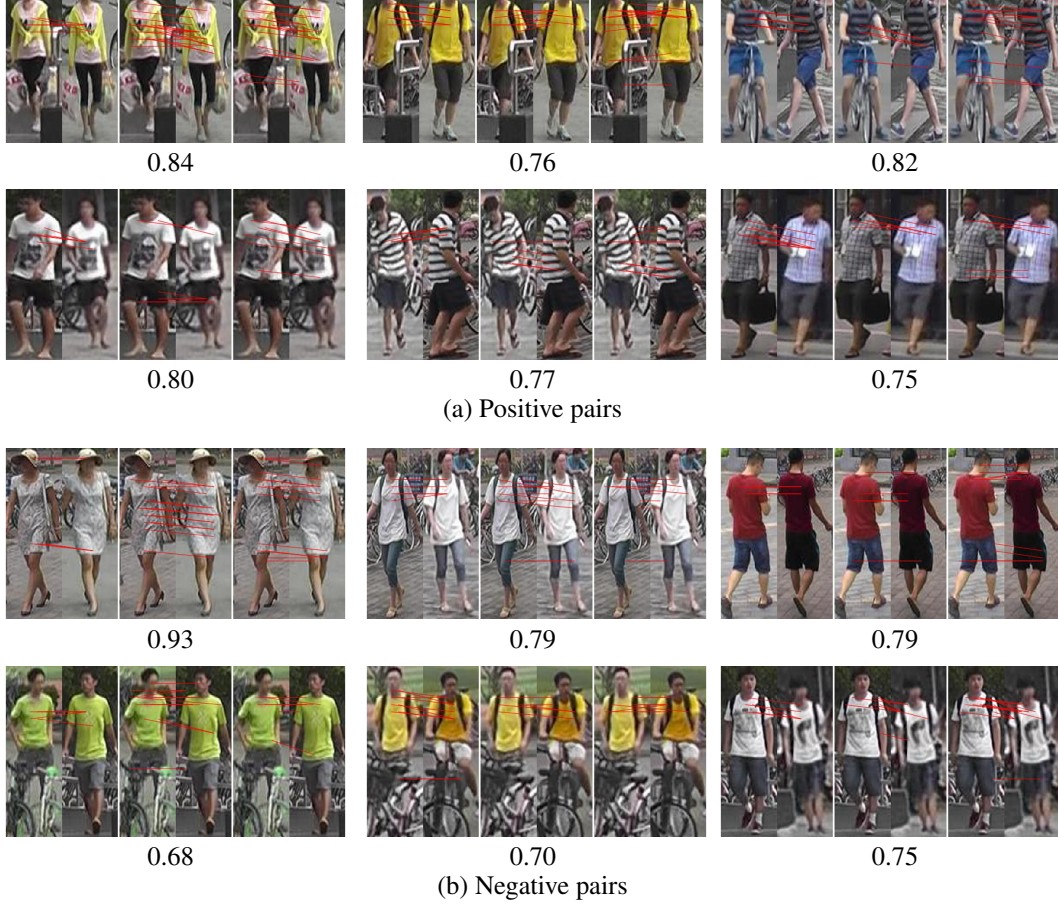

Figure 3: Examples of qualitative matching results on the Market-1501 dataset, by the proposed TransMatcher trained on the MSMT17 dataset. For each pair of images, local correspondence matches found on the three layers of the TransMatcher are shown. Numbers represent similarity scores.

# 7 Conclusion

With the study conducted in this paper, we conclude that: (1) direct applications of ViT and the vanilla Transformer are not effective for image matching and metric learning, because they lack cross-image interaction in their designs; (2) designing query-gallery concatenation in ViT does not help, while introducing query-gallery cross-attention in the vanilla Transformer leads to notable but not adequate improvements, probably because the attention mechanism in Transformers might be primarily designed for global feature aggregation, which is not naturally suitable for image matching; and (3) a new simplified decoder thus developed, which employs hard attention to cross-matching similarity scores, is more efficient and effective for image matching and metric learning. With generalizable person re-identification experiments, the proposed TransMatcher is shown to achieve state-of-the-art performance on several popular datasets with large improvements. Therefore, this study proves that Transformers can be effectively adapted for the image matching and metric learning tasks, and so other potentially useful variants will be of future interest.

## Acknowledgements

The authors would like to thank Yanan Wang who helped producing Fig. 1 in this paper, and Anna Hennig who helped proofreading the paper, and all the anonymous reviewers for the valuable feedbacks in improving the paper.

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
