# TransMatcher: Deep Image Matching Through Transformers for Generalizable Person Re-identification: Appendix

**Shengcai Liao**[*] **and Ling Shao**
Inception Institute of Artificial Intelligence (IIAI), Abu Dhabi, UAE
https://liaosc.wordpress.com/

## A  mAcc Measure

For ease and reliable comparison, we report the average of all Rank-1 and mAP results on all test datasets over several random runs for ablation study and parameter analysis. This is denoted by mAcc. There are three reasons that we use mAcc.

- It is a unified measure, which is convenient for algorithm comparison. Both Rank-1 and mAP are accuracy measures ranging from 0%-100%, thus averaging them is possible. Besides, if a method's mAcc is 1% higher than another method, on average it means that every single measure on each dataset has been increased by 1%, which is a perceptible achievement.

- Some algorithms perform unstably across different runs, thus the average among several runs is a more stable measure.

- Using a unified measure is convenient, concise, and space-saving for ablation study and parameter analysis.

## B  Additional Description and Analysis of the Proposed Method

### B.1  Symmetric Extension of GMP

In Eq. (6) of the main paper, $S'_n$ is of size $HW \times hw$, with the first $HW$ associated with the query feature map $Q$, and the second $hw$ associated with the gallery feature map $K$. Here $H = h$ and $W = w$, but to be clear, let's denote them differently. Then in Eq. (7), GMP is applied along the last dimension of $hw$ elements, resulting in a vector of size $HW$. However, considering that for Q and K their role and order are exchangeable, we can have Eq. (7') that $S''_n = \max(S'_n, \text{dim=0})$, that is, applying GMP along the first dimension of $HW$ elements, resulting in a vector of size $hw$. Afterwards, the two sets of results are independently processed by the MLP head and the final scores are summed. This way, if the input pair of images are swapped, the final similarity score will remain the same. Please refer to the source code of TransMatcher [2] for further details.

### B.2  Prior Score Embeddings

The prior score embeddings are learnable parameters of size $hw \times hw$. They can also be considered learnable weights, somewhat similar to the learnable FC weights. They act as spatial matching priors. For example, for a pair of person images to be matched, one head should be matched to the other head, and so the corresponding rough "head-to-head" locations in the $hw \times hw$ parameters should have large values, while others such as "head-to-foot" locations should have small values. It is not

---

[*]Shengcai Liao is the corresponding author.
[2]https://github.com/ShengcaiLiao/QAConv/tree/master/projects/transmatcher

35th Conference on Neural Information Processing Systems (NeurIPS 2021).

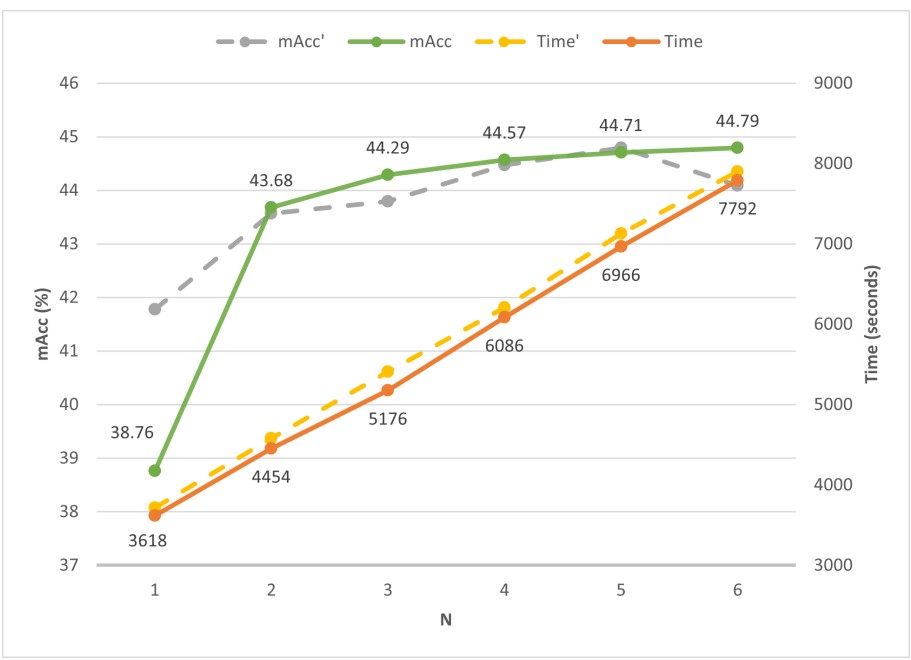

Figure A: Performance of different layer configurations of the proposed TransMatcher. With the same number of decoders, mAcc' and Time' are results of the configuration where the same number of encoders are paired with the decoders, while mAcc and Time are results of the configuration where the first encoder is replaced directly by the output of the deep feature map, which is the preferred default configuration.

easy to define this manually, so we make it learnable so that this prior can be automatically learned from data. Furthermore, this can also be understood as the image matching extension of the original positional embedding proposed in Transformers.

### B.3 Different Layer Configurations of TransMatcher

As mentioned in the main paper, in practice, we find that replacing the first encoder directly with the output of the deep feature map slightly improves the results while being more efficient. Specifically, if $N$ decoder layers are used, normally there should be $N$ corresponding encoder layers, as shown in Fig. 1 of the main paper. Then, to save some computation, we only use $N-1$ encoder layers, together with the CNN feature map as the first layer to pair the decoders. For these two configurations, the experimental results are illustrated in Fig. A, where it can be seen that replacing the first encoder slightly improves the running speed, while at the same time slightly improves the accuracy, except when there is only one decoder. However, note that when there is only one decoder, there is no encoder for the default configuration since the first encoder is replaced directly by the CNN feature map. Therefore, in this case the other configuration is better, which also proves the benefit of including an encoder.

### B.4 Encoder V.S. Decoder

Encoders are indeed useful, but not as important as the proposed decoder. This can be understood from the experiments. ViT is a pure encoder based model, without decoders. However, from Table 2 of the main paper we can see that its generalization performance is not satisfactory. In contrast, Fig. A in this Appendix provides an example where $N=1$ corresponds to a model with only decoders but no encoders. In this case the mAcc is 38.76%, which is much better than the 27.42% of ViT in Table 2 of the main paper. The reasons may be two folds. Firstly, with only encoders, though it is useful for feature learning, we observe that it may easily overfit the source data. Secondly, inspired by QAConv [11], the decoders perform local image matching explicitly, which is interpretable and has a better generalization performance.

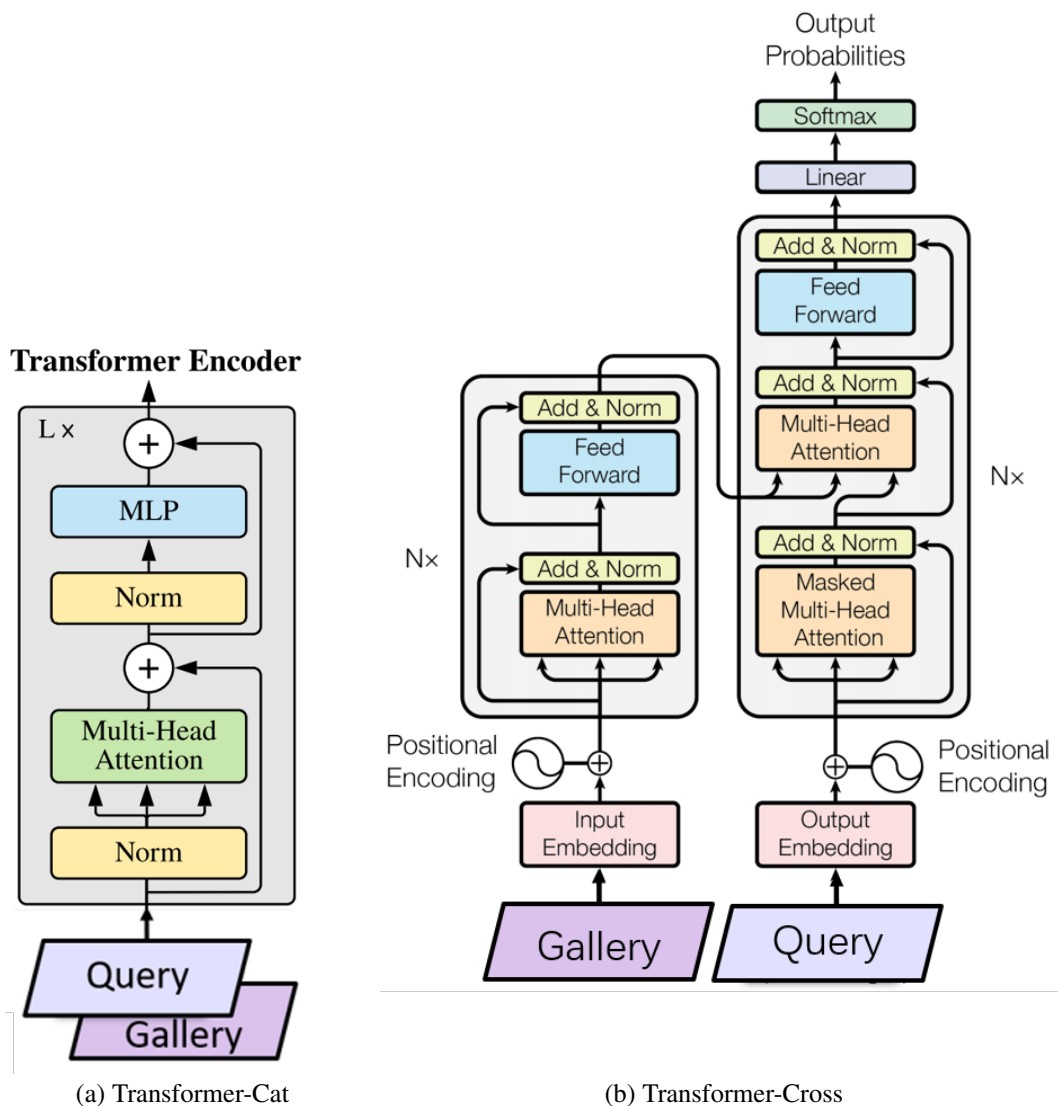

(a) Transformer-Cat               (b) Transformer-Cross

Figure B: Illustration of (a) Transformer-Cat (adapted from [8]) and (b) Transformer-Cross (adapted from [16]).

### B.5 Efficiency And Practical Value

First, this work has theoretical value for understanding Transformer's capability in image matching. Second, generalizable person re-identification is particularly designed towards practical applications, and as shown in Table 1 of the main paper, the proposed method has large improvements over existing methods, therefore, it deserves further study, e.g. improving its efficiency. Third, the proposed method has already considered the efficiency, with its simplified decoder and balanced parameter selection, and thus it is the most efficient one in cross-matching Transformers as shown in Table 2 of the main paper. Other potential Transformers would encounter more difficulty in efficiency. Finally, compared to the SOTA method QAConv-GS which spends 0.96 hour in training MSMT17, the proposed method spends 1.44 hours, which is still acceptable.

## C Implementation Details And Variants of Different Transformers

Illustration of Transformer-Cat and Transformer-Cross is shown in Fig. B. The same binary cross-entropy loss in GS sampler [12] is applied for all methods.

## C.1 ViT

ViT only contains Transformer encoders, and the same vanilla Transformer encoders are used for both ViT and the proposed method. The MLP head for the proposed model is only used in the decoder. Besides, the structure of the MLP head is almost the same as the feed-forward layer in the ViT encoder, where two FCs are used, together with ReLU and normalization.

## C.2 Transformer-Cross

For Transformer-Cross, as in vanilla Transformers, both $K$ and $V$ are the same and they are both from the encoded memory (gallery features). In Transformer-Cross, encoders are applied only for gallery features but we input query features to decoders directly without encoders, this is because:

- Almost all components in Transformer encoders are already designed in decoders, such as the self-attention module (prior to cross-attention), the feed-forward layer, etc.

- Besides, this is also to be consistent to existing Transformers (e.g. vanilla Transformer and DeTR). We did not see a method inputting query tokens to encoders before decoders.

- For the proposed method, the same encoders are applied for both gallery and query features. This is because, first, we think query and gallery are exchangeable and so we would like to design a symmetric distance metric. Second, the proposed decoder is simplified, without a self-attention module as in the vanilla Transformer decoder.

For a variant, we further apply the encoders to query features prior to decoders for Transformer-Cross. The mAcc for this variant is 33.77%, which is lower than 36.70% reported in Table 2 of the main paper. This may be because this structure is too complex to learn an effective distance metric.

## C.3 Fusion in Transformer-Cat

For Transformer-Cat, we also tried appending the results of the Transformer in the lower block. At first, we tried score fusion as in the proposed method, resulting in an mAcc of 24.92%, compared to 25.34% in Table 2 of the main paper. Then we thought maybe the final normalization layer in the default encoder hindered the improvement; after removing it we got 25.10%. Later, we tried feature fusion instead of score fusion, then we got 25.22%. In any case, we were not able to improve the results by multi-layer fusion. This may be because the Transformer-Cat structure itself is not suitable for metric learning, as explained in Section 5 in the main paper.

## C.4 Improved Components on Transformer-Cross

As in ViT and DeTR, we would like to see what's the capability of the original Transformers for distance metric learning. Therefore, we did as little modifications as possible to the baseline architectures. Besides, both Transformer-Cat and Transformer-Cross are heavy to compute, which limits their values and further developments. However, we still tried several variants, including multi-scale fusion among different Transformer layers, and shared FC for $Q$ and $K$ in cross-attention, to see the maximal capability of Transformer-Cross.

- For multi-scale fusion, the mAcc is 29.52%, lower than the 36.70% in Table 2 of the main paper, and we observe that the learning is not stable across different runs. Again, this may be because the structure of Transformer-Cross does not directly target on image matching and metric learning.

- An interesting finding is the role of shared FC for $Q$ and $K$ in cross-attention in decoders. Previously, we set shared FC in the proposed model just because the distance metric is required to be symmetric. We did not observe and did not expect it to be critical in the proposed pipeline (see Table 3 of the main paper for ablation study). On the other hand, Transformer-Cross is already not a symmetric design for gallery-query pairs, therefore, two different FCs are reasonable, as in the vanilla Transformer. However, now when we force the FC to be shared on both $Q$ and $K$ in Transformer-Cross, surprisingly, the mAcc becomes 41.56%, which is much better than 36.70% in Table 2 of the main paper, and it is only slightly worse than the 42.12% of the proposed method. We guess this is because forcing

shared FC makes the feature space of $Q$ and $K$ being consistent before cross-attention, and therefore it helps the subsequent metric learning task. Nevertheless, compared to the proposed model, the Transformer-Cross is still too heavy, costing too much memory and computation time.

- Based on the shared FC, we tried the multi-scale fusion again, and got 41.52% for mAcc.

## D  Comparison to Other Methods

### D.1  Comparison to DeTR

DeTR [1] is an original work for Transformer based object detection. Though DeTR is not directly applicable to person re-identification due to its ad-hoc detection-oriented structure, we can see that beyond the particular detection head design, DeTR is very similar to the vanilla Transformer compared in this paper, except that to adapt to the person re-identification task (pairwise metric learning), our prediction head outputs pairwise similarities between gallery-query pairs. Therefore, the vanilla Transformer compared in this paper can be considered the person re-identification version of DeTR. Furthermore, there are learnable queries in DeTR, which inspires us that, how about using actual image queries instead of learnable queries? This results in the Transformer-Cross method proposed in the paper, which is also compared in the experiments.

### D.2  Why Not Other Attention Modules?

Other attention modules, such as the Non-local Network [17] and the channel-wise attention [6, 19], are mostly for feature representation learning, particularly, feature enhancement or feature refinement, which operates within the same image. Please refer to the Related Work section of the main paper regarding this. However, image matching or distance metric learning is a different task, involving pairs of images. On the other hand, as motivated in the paper, Transformers with almost its original form have shown great success on computer vision tasks recently (e.g. ViT, DeTR), and image matching is also a typical computer vision task (e.g. face recognition and person re-identification), therefore, this is a timely study that whether Transformers are useful for image matching and how to apply or adapt them for this task. Furthermore, there are learnable queries in DeTR, which inspires us that, how about using actual image queries instead of learnable queries? This results in the Transformer-Cross method proposed in the paper.

### D.3  Comparison to ResNet-IBN And FastReID

ResNet-IBN has already been compared in Table A of the supplementary material, where the DualNorm method is a straightforward extension of ResNet-IBN, and the proposed method performs much better than it. FastReID is a strong baseline for person re-identification, and we find some results from `https://github.com/JDAI-CV/fast-reid/tree/master/projects/CrossDomainReID`, where the results are mostly with the DukeMTMC-reID dataset which has been officially removed due to some ethic concerns. Yet there is a result for training on Market1501 while testing on MSMT17, with Rank-1 29.8% and mAP 10.3%. Compared to the results in Table 1 of the main paper, the proposed method performs much better, with Rank-1 47.3% and mAP 18.4%.

## E  Additional Comparison to The State of The Art

### E.1  Datasets

Some recent works, such as DIMN [14], DualNorm [7], and DDAN [2], used different experimental settings. To compare with them, we conducted additional experiments following their experimental protocols. Specifically, a large-scale combined source training dataset is constructed, which includes the CUHK02 [9], CUHK03 [10], Market-1501 [21], DukeMTMC-reID[3] [3, 23], and CUHK-SYSU Person Search [20] datasets. Note that all the images in these datasets are used for training, regardless

---

[3]Note that the DukeMTMC and its derived datasets have been officially removed due to some ethic concerns. Here we include it only for the sake of comparison to some existing results. We discourage further usage of DukeMTMC datasets in the future.

of their original training and test subset splits. This results in a large-scale training data, including 18,530 identities and 121,765 training images.

As for testing, four small datasets are used, including the VIPeR [4], PRID [5], QMUL GRID [13], and i-LIDS [22] datasets. The standard testing splits of these datasets are used for evaluation. Specifically, 10 random splits of training and test subsets for each dataset are repeated for evaluation, with the averaged results reported. The single-shot evaluation protocol is used for all experiments. For each split, the probe/gallery splits of image subsets are as follows: VIPeR: 316/316; PRID: 100/649; GRID: 125/900; and i-LIDS: 60/60. Besides, on VIPeR, their swapped versions of the probe/gallery splits are also evaluated and averaged.

Table A: Comparison of the state-of-the-art direct cross-dataset evaluation results (%). Mob is short for MobileNetV2. Res is short for ResNet-50. DN is short for DualNorm [7].

| Method | Venue | Net | VIPeR | | PRID | | GRID | | i-LIDS | | Average | |
|---|---|---|---|---|---|---|---|---|---|---|---|---|
| | | | R1 | mAP | R1 | mAP | R1 | mAP | R1 | mAP | R1 | mAP |
| DIMN [14] | CVPR'19 | Mob | 51.2 | 60.1 | 39.2 | 52.0 | 29.3 | 41.1 | 70.2 | 78.4 | 47.5 | 57.9 |
| DualNorm [7, 2] | BMVC'19 | | 53.9 | 58.0 | 60.4 | 64.9 | 41.4 | 45.7 | 74.8 | 78.5 | 57.6 | 61.8 |
| BCaR[15] | BMVC'20 | | 50.4 | - | 37.1 | - | 31.9 | - | 68.7 | - | 47.0 | - |
| BCaR + DN[15] | BMVC'20 | | **57.3** | - | 62.0 | - | 42.3 | - | **80.0** | - | 60.4 | - |
| DDAN [2] | AAAI'21 | | 52.3 | 56.4 | 54.5 | 58.9 | **50.6** | 55.7 | 78.5 | 81.5 | 59.0 | 63.1 |
| DDAN + DN [2] | AAAI'21 | | 56.5 | 60.8 | 62.9 | 67.5 | 46.2 | 50.9 | 78.0 | 81.2 | 60.9 | 65.1 |
| QAConv-GS [12] | arXiv'21 | | 47.6 | 57.2 | 61.3 | 68.8 | 37.4 | 45.3 | 75.7 | 82.3 | 55.5 | 63.4 |
| TransMatcher | Ours | | 53.1 | **63.1** | 65.6 | **74.3** | 48.8 | **56.4** | 77.8 | **84.2** | 61.3 | 69.5 |
| DualNorm[7, 2] | BMVC'19 | Res | 59.4 | - | 69.6 | - | 43.7 | - | 78.2 | - | 62.7 | - |
| BCaR + DN[15] | BMVC'20 | | **65.8** | - | **70.2** | - | 52.8 | - | 81.3 | - | **67.5** | - |
| QAConv-GS [12] | arXiv'21 | | 57.8 | 67.5 | 63.0 | 71.5 | 51.9 | 61.3 | 79.2 | 85.4 | 63.0 | 71.4 |
| TransMatcher | Ours | | 63.4 | **71.8** | 63.8 | **72.0** | **57.2** | **65.7** | **81.8** | **87.8** | 66.6 | **74.3** |

### E.2 Results

For a fair comparison to the existing results, MobileNetV2 with width multiplier of 1.0 is additionally used for QAConv-GS and TransMatcher as the backbone network. The same configuration and training setting is applied as described in the main paper, except that the learning rates are decayed by 0.1 after 6 epochs, and 9 epochs are trained in total, since with the GS sampler the number of iterations per epoch is determined by the number of classes and the combined source training dataset has 18,530 identities. In addition, gradient clipping [12] is applied with $T = 32$ for MobileNetV2 other than $T = 4$ with ResNet-50, because MobileNetV2 is small and less suffered from overfitting.

The evaluation results are shown in Table A. From the results it can be observed that the proposed TransMatcher with the ResNet-50 backbone outperforms QAConv-GS with a large margin. With the MobileNetV2 backbone, the proposed TransMatcher also achieves the best results on average, though on some datasets it has slightly lower Rank-1 results. This shows TransMatcher's good learning capability from large-scale combined datasets. Besides, it can be seen that methods with the ResNet-50 backbone are much better than the MobileNetV2, though MobileNetV2 is theoretically with less parameters and computational costs. However, running on GPU cards, we find that MobileNetV2 is not that efficient, which requires 153,903 seconds for training with TransMatcher on the combined training dataset. In contrast, TransMatcher with ResNet-50 requires 169,257 seconds for training, while it is 137,511 seconds for ResNet-18.

## F  Qualitative Analysis

With the help of the GMP layer, inspired from QAConv [11], the proposed TransMatcher is able to find the best local correspondence matches in each decoder layer. Some qualitative matching results are shown in Fig. C for a better understanding of the proposed method. The model used here is trained on the MSMT17 dataset [18], and the evaluations are done on the query subsets of the Market-1501 dataset [21]. Results of both positive pairs and hard negative pairs are shown. For a clear illustration, only reliable correspondences with matching scores over a certain threshold are shown, where the

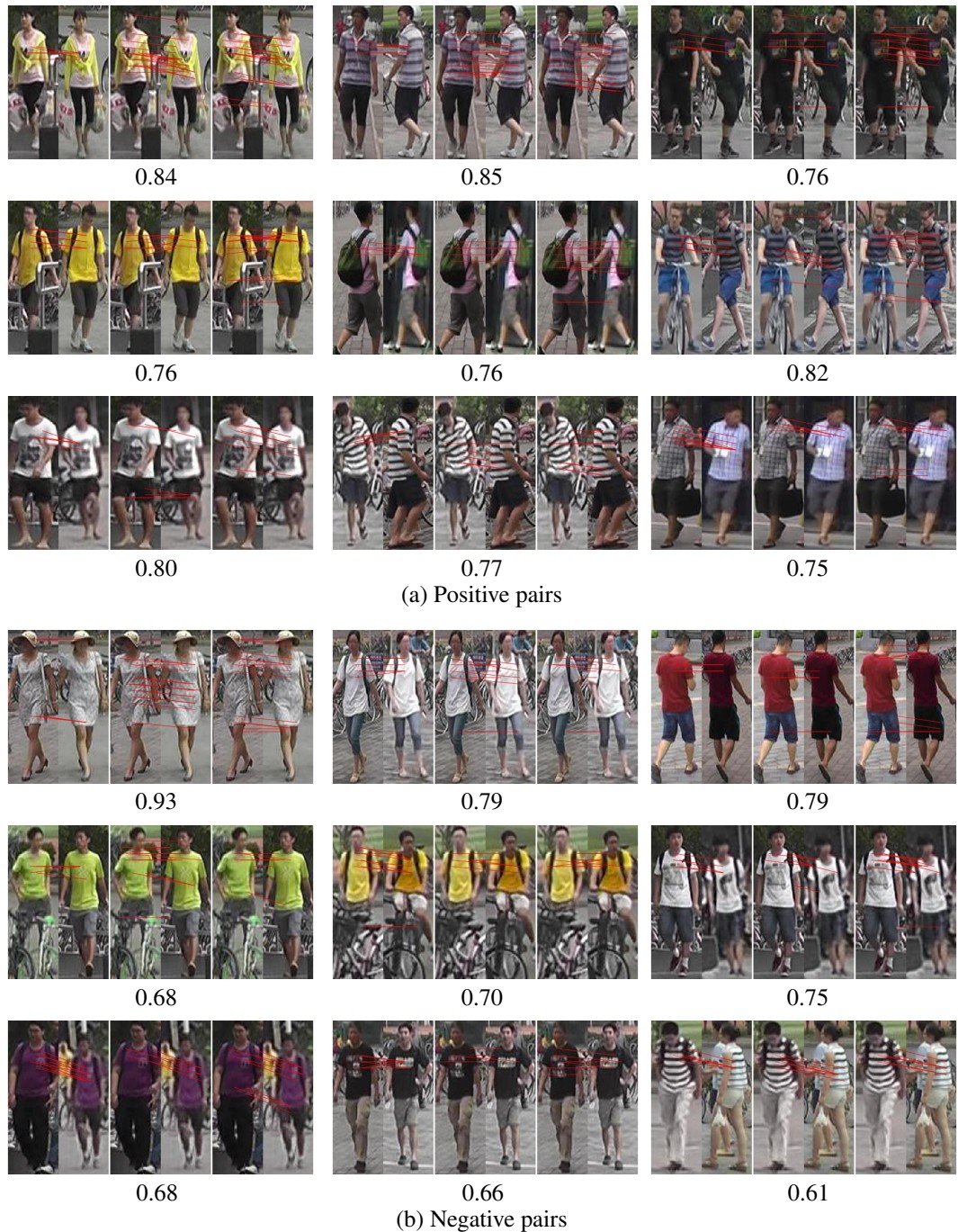

0.84     0.85     0.76

0.76     0.76     0.82

0.80     0.77     0.75

(a) Positive pairs

0.93     0.79     0.79

0.68     0.70     0.75

0.68     0.66     0.61

(b) Negative pairs

Figure C: Examples of qualitative matching results on the Market-1501 dataset, by the proposed TransMatcher using the model trained on the MSMT17 dataset. For each pair of images, local correspondence matches found on the three layers of the TransMatcher are shown. Numbers represent similarity scores.

threshold is determined by a false acceptance rate of 1‰ over all matches of negative pairs. Note that the local positions are coarse due to the $24 \times 8$ size of the feature map.

As can be observed from Fig. C, the proposed method is able to find correct local correspondences for positive pairs of images, even if there are notable misalignments in both scales and positions, pose, viewpoint, and illumination variations, occlusions, and low resolution blur. Besides, for hard negative pairs, the matching of TransMatcher still appears to be mostly reasonable, by linking visually similar parts or even the same person who might be incorrectly labeled.

This indicates that the proposed TransMatcher is effective in local correspondence matching, and note that it learns to do this with the only supervision of identity information. Besides, the matching capability is generalizable to other datasets beyond the training set. From the illustration it can also be seen that, generally, matching results of the first decoder layer are not as successful as the next two layers, and the matching with the last decoder layer appears to be the best. This indicates that both Transformer encoders and decoders helps the model to match better by aggregating global similarity information.