# OpenReview forum: "TransMatcher: Deep Image Matching Through Transformers for Generalizable Person Re-identification"
_NeurIPS.cc/2021/Conference — NeurIPS 2021 Poster_

### Official Review · Reviewer_XZ51 · 2021-07-16

**Rating:** 6
**Confidence:** 5

**Summary:**

This paper proposes a Transformer-Based Deep Image Matching approach for generalizable person re-identification. It adopts two simple solutions based on the transformer to achieve pair image matching. The idea is simple and easy to follow.

**Limitations And Societal Impact:**

There are four limitations:
1.	In this experiment, single dataset training and single dataset testing cannot verify the generalizable ability of models, it should conduct experiments on large-scale datasets.
2.	The efficiency of such pairwise matching is very low, making it difficult to be used in practical application systems.
3.	I hope to see that you can compare your model with ResNet-IBN / ResNet of FastReID, which is practical work in the person Reid task.
4.	I think the authors only use the transformer to achieve the local matching, therefore, the contribution is limited.



**Main Review:**

This paper proposes a Transformer-based image matching to achieve generalizable person re-identification.

This idea is based on the assumption that local feature matching helps to improve the generalization ability of models. I agree with this conclusion. The cross image attention proposed in this paper is indeed making sense for image-to-image matching. The ablation studies and experiments are convincing and demonstrate the improvement of the proposed TansMatcher.


However, the authors only use the transformer to achieve the local matching, therefore, the contribution is limited.


**Time Spent Reviewing:**

4

---

> ### Author Response · Authors · 2021-08-10
> **Author Feedback**
>
> Dear XZ51,
>
>
>
> Thanks for your time and effort. We appreciate the positive rating and all the valuable comments. In the following we shall address your concerns point by point.
>
>
>
> # Limited Contribution
>
> Hi, we are sorry but we do not quite understand what the actual meaning is for this judgement. Do you mean Transformers are powerful and should be applied in a more sophisticated way? If our understanding is correct, the two original baselines (ViT and the vanilla Transformer) and the two naive solutions we proposed (Transformer-Cat and Transformer-Cross) are all comprehensive models following the original Transformer. However, they are not satisfactory for our task, except that Transformer-Cross we proposed is promising. You are right, Transformers are powerful, and by its design its capability is automatically learned from data. However, in this study we find that for the pairwise image matching or distance metric learning task, the original design of Transformers is not straightforward for this task, and so the baseline models encounter some difficulties in metric learning from pairs of data. This is why we need to modify its structure for the image matching task. Accordingly, we design a model which is successful. It is simple, but efficient, and straightforward and effective for image matching. There may be other possibilities by integrating self-attention and cross-attention in sophisticated ways, but there are also uncertainties in their computation and memory loads, and effectiveness in addressing image matching.
>
> We are not sure whether the above understanding is correct. If we misunderstand your question, please do let us know in the discussion phase so that we have the chance to further address your concern.
>
>
>
> #  Large-scale Datasets
>
> In the experiments, both MSMT17 (4,101 IDs) and RandPerson (8,000 IDs) are already the current largest person re-identification datasets in real-world and virtual data. Besides, the testing are performed on multiple datasets. Furthermore, in the supplementary material, we do provide experimental results by training on a very large training data (18,530 IDs) composed by a number of existing datasets, and testing on four popular datasets, where the proposed method is also shown to be superior to existing methods.
>
>
>
> # Efficiency And Practical Value
>
> First, this work has theoretical value for understanding Transformer's capability in image matching. Second, generalizable person re-identification is particularly designed towards practical applications, and as shown in Table 1, the proposed method has large improvements over existing methods, therefore, it deserves further study, e.g. improving its efficiency. Third, the proposed method has already considered the efficiency, with its simplified decoder and balanced parameter selection, and thus it is the most efficient one in cross-matching Transformers as shown in Table 2. If this is not valuable, other potential Transformers would encounter more difficulty in efficiency. Finally, compared to the SOTA method QAConv-GS which spends 0.96 hours in training MSMT17, the proposed method spends 1.44 hours, which is still acceptable.
>
>
>
> # Comparison to ResNet-IBN And FastReID
>
> Thanks for the suggestion. ResNet-IBN has already been compared in Table 1 of the supplementary material, where the DualNorm method is a straightforward extension of ResNet-IBN, and the proposed method performs much better than it. For FastReID, indeed it is a very strong baseline, and we find some results from https://github.com/JDAI-CV/fast-reid/tree/master/projects/CrossDomainReID, where the results are mostly with the DukeMTMC-reID dataset which has been officially removed due to some concerns. Yet there is a result for Market1501 -> MSMT17, with Rank-1 29.8% and mAP 10.3%. Compared to the results in Table 1, the proposed method performs much better, with Rank-1 47.3% and mAP 18.4%. We will include this in Table 1.
>
>
>
> # Summary
>
> Hope the above have addressed your concerns. We will make necessary revisions to the paper according to the above discussions. Thanks again for your valuable comments.

---

### Official Review · Reviewer_Gj64 · 2021-07-16

**Rating:** 6
**Confidence:** 4

**Summary:**

This paper investigates how to apply the transformer architecture to solve the image match problem of generalised person Re-ID. The features of query/gallery images are first extracted with a ResNet model, then a transformer encoder without positional embedding is applied to query/gallery features, respectively. A customised decoder is designed to obtain the final match score for the following pairwise metric learning. The experiments are conducted on several common Re-ID datasets and the proposed method have shown better performance than the baseline.

**Limitations And Societal Impact:**

Yes, I think so.

**Main Review:**

STRENGTH
+ The performance on CUHK03/Market-1501/MSMT17 is superior to the baseline methods.
+ The idea of applying the transformer model to person Re-ID seems to be new.
+ The paper writing is easy to follow.

WEAKNESS
- There are other attention modules, e.g. no-local or channel-wise attention, why particularly use the transformer if no positional embedding is required?
- The proposed pipeline is still a naive solution. TThe transformer encoder is simply used to refine the ResNet features, while the query /gallery features are combined together through a simple dot product in decoder. I don't think the cross attention between query/gallery can be learned in this way - it is more likely the performance improvement is gained by the transformer encoder instead of the proposed decoder.
- The comparison with ViT and transformer in Table 2 is not quite fair, since the authors use a smaller transformer model (line 263-267) due to memory overflow problem. The original settings may actually work better
- Since both person-ID and self-attention have the same concept "query" but having quite different meanings, the authors should make clear which query is referred to in the paper to avoid confusion.
- Intuitive visualizations can be helpful to reveal the reason why the transformer can work better for person-ReID problems.

Overall, although this paper has reported higher performance than the baseline, the motivation of using transformer is not very clear, the pipeline design is naive and lack novelties, and some experimental results are still questionable. Considering the superior performance, I vote for a marginally below.

------After rebuttal------
The authors solved parts of my concerns and therefore I increase the rating to marginally above (6).

**Time Spent Reviewing:**

4

---

> ### Author Response · Authors · 2021-08-10
> **Author Feedback**
>
> Dear Gj64,
>
>
>
> Thanks for the positive comments and other valuable feedbacks. We shall address your concerns point by point as follows.
>
>
>
> # Other Attention Modules
>
> Other attention modules, such as the Non-local Network and the channel-wise attention, are mostly for feature representation learning, particularly, feature enhancement or feature refinement, which operates within the same image. However, image matching or distance metric learning is a different task, involving pairs of images. On the other hand, as motivated in the paper, Transformers with almost its original form have shown great success on computer vision tasks recently (e.g. ViT, DeTR), and image matching is also a typical computer vision task (e.g. face recognition and person re-identification), therefore, this is a timely study that whether Transformers are useful for image matching and how to apply or adapt them for this task. Furthermore, there are learnable queries in DeTR, which inspires us that, how about using actual image queries instead of learnable queries? This results in the Transformer-Cross method proposed in the paper.
>
>
>
> # Encoder V.S. Decoder
>
> No, we do not agree. Encoders are indeed useful, but not as important as the proposed decoder. This can be understood from the experiments. ViT is a pure encoder based model, without decoders. However, from Table 2 we can see that its generalization performance is not satisfactory. In contrast, Section B and Fig. 1 in the supplementary material provides an example where N=1 corresponds to a model with only decoders but no encoders. In this case the mAcc is 38.76%, which is much better than the 27.42% of ViT in Table 2. The reasons may be two folds. Firstly, with only encoders, though it is useful for feature learning, we observe that it may easily overfit the source data. Secondly, the decoders perform local image matching explicitly, which is interpretable and has a better generalization performance.
>
>
>
> # Unfair Small Transformer Models
>
> Well, we have used a GPU (NVIDIA V100) with almost the largest GPU memory (32GB). If even a small Transformer model (e.g. three layers) makes the 32GB memory overflow and takes a long time to compute, how can we expect large model to be accepted by researchers in practice? For example, Reviewer XZ51 also has an interest in efficiency. While we are not able to have an exact answer whether bigger Transformer-Cat or Transformer-Cross in Table 2 will perform better, we are sure of two things. First, the comparison in Table 2 is fair, with the same scale of configurations. Second, reducing the computation and memory cost of Transformers is important and valuable in practice, as done in the proposed decoder.
>
> Furthermore, getting higher accuracy with larger model is not our pursuit in this paper. As can be seen from Fig. 2, we are able to get higher accuracy (other than the default mAcc=44.29%) with larger $d$ (default $d=512$) and $N$ (default $N=3$), however, at the same time we care much about the training time and always make a balance (Section 6.6), and we only report the results of $d=512$ and $N=3$ in Table 1 for the SOTA comparison and in Table 2 for Transformer comparison.
>
>
>
> # The Query Concept
>
> Thanks for pointing out this. Indeed they are the same word "query" but from two different domains. However, in concept they have the same meaning, that is, giving an input data, request the output by similarity comparison from the stored data (memory or gallery). They both have this concept which is in line with information retrieval: person re-identification is a typical image retrieval task, and the notations of Q, K, and V in Transformers are also from information retrieval. This is also why we make a link of them and propose the Transformer-Cross method. But you are right, they have different form in different stages. We will make necessary clarifications to avoid confusion.
>
>
>
> # Intuitive Visualizations
>
> Thanks for the suggestion. We do provide some visualizations and qualitative analysis, in Section C and Fig. 2 in the supplementary material. The illustration indicates that the proposed TransMatcher is effective in local correspondence matching, and this capability is more generalizable to unseen domains, for example, to address unknown misalignments. We will try to include some of these illustrations in the main text if possible.
>
>
>
> # The Pipeline Design Is Naive And Lack Novelties
>
> No, we completely do not agree. As explained in the above, the proposed decoder is more important than the vanilla encoders, and the simplified design helps a lot in reducing the computation and memory cost. Furthermore, this simplifed structure is still able to learn cross attentions between gallery and query images, as explained in the intuitive visualizations. Instead of "naive", we would rather say it is simple but effective. Why not a concise structure be preferred if it is superior in performance? For the lacking novelty judgement, could you help show us some citations doing similar works? Thanks. We did not find a similar method as the proposed TransMatcher, and we did not find a study discussing Transformers' ability for generalizable person re-identification in the form of pairwise image matching and metric learning.
>
>
>
> # Summary
>
> Hope the above have addressed your concerns on motivation, novelty, and unfair experiments. We will make necessary revisions to the paper according to the above discussions. In summary, this paper provides a new view, from the point of image matching and metric learning requiring pairs of images (which is quite different from other popular tasks, such as image classification and object detection which only involve a single image), on how to understand Transformers, what's its capability and limitations on such task, and how to efficiently adapt it, with faster speed and less memory consumption. We hope the above weaknesses have been addressed and the rating can be re-evaluated. Thanks.

---

### Official Review · Reviewer_SVbe · 2021-07-17

**Rating:** 6
**Confidence:** 4

**Summary:**

This paper studies person re-identification using visual transformers with query and gallery interactions. The paper presents first simple query and gallery image concatenation and cross attention layer for interaction query and gallery interactions; and later presents the proposed TransMatcher architecture, which replaces the decoder self-attention by only keeping query-key matching followed by global max pooling and multi-layer perceptron head maps composed of fully connected and batch norm layers.


**Limitations And Societal Impact:**

It mentions the limitations and possible societal impact.

**Main Review:**

The analysis on the standard transformer for detection is interesting. I further appreciated that it presents the naive extensions that do not lead to improvement along with the proposed model that leads to improvement.

The visual transformers is an active area of research and most of the very recent (a few months) papers are also cited in the related work.
The experimental setup contains comparisons between the proposed approach and several earlier studies on the re-identification task. However, it would be further interesting to see how the proposed model performs compared to DeTR which is known for object detection on the re-identification task.

It is not very clear how the global max pooling is symmetrized, could you please provide the equations of this part and the concatenation afterwards, what is the dimension of the output?

How are the prior score embeddings obtained? It is not clear why it is used, what is the rationale behind it?

Sharing the code will increase the reproducibility of results. Could you please share your implementation?


---
Line 170, 171, 173, 177 $\mathbb{R}$ instead of $R$ for the dimensions


**Time Spent Reviewing:**

4-5 hours

---

> ### Author Response · Authors · 2021-08-10
> **Author Feedback**
>
> Dear SVbe,
>
>
>
> Thanks for your time and effort. We appreciate the positive rating and all the valuable comments. In the following we shall address your concerns point by point.
>
>
> # Comparison to DeTR
>
> Thanks for the suggestion. You are right, DeTR is a great work and the earliest work for Transformers based object detection. We have learned a lot from DeTR. Though DeTR is not directly applicable to person re-identification due to its ad-hoc detection-oriented structure, we can see that beyond the particular detection head design, DeTR is very similar to the vanilla Transformer compared in this paper, except that to adapt to the person re-identification task (pairwise metric learning), our prediction head outputs pairwise similarities between gallery-query pairs. Therefore, the vanilla Transformer compared in this paper can be considered the person re-identification version of DeTR. Furthermore, there are learnable queries in DeTR, which inspires us that, how about using actual image queries instead of learnable queries? This results in the Transformer-Cross method proposed in the paper, which is also compared in the experiments.
>
>
>
> # Symmetric Extension of GMP
>
> Sorry for the unclear description. In Eq. (6), $S'_n$ is of size $HW \times hw$, with the first $HW$ associated with the query feature map $Q$, and the second $hw$ associated with the gallery feature map $K$. Here $H=h$ and $W=w$, but to be clear, let's denote them differently. Then in Eq. (7), GMP is applied along the last dimension of $hw$ elements, resulting in a vector of size $HW$. However, considering that for Q and K their role and order are exchangeable, we can have Eq. (7') that $S{''}_n = \text{max}(S'_n, \text{dim=0})$, that is, applying GMP along the first dimension of $HW$ elements, resulting in a vector of size $hw$. Afterwards, the two sets of results are independently processed by the MLP head and the final scores are summed. This way, if the input pair of images are swapped, the final similarity score will remain the same.
>
>
>
> # Prior Score Embeddings
>
> Sorry for the unclear description. The prior score embeddings are learnable parameters of size $hw \times hw$. They can also be considered learnable weights, somewhat similar to the learnable FC weights. They act as spatial matching priors. For example, for a pair of person images to be matched, one head should be matched to the other head, and so the corresponding rough "head-to-head" locations in the $hw \times hw$ parameters should have large values, while others such as "head-to-foot" locations should have small values. It is not easy to define this manually, so we make it learnable so that this prior can be automatically learned from data. Furthermore, this can also be understood as the image matching extension of the original positional embedding proposed in Transformers.
>
>
>
> # Sharing the Code
>
> Sure, we are more than happy to share the code, as promised in the abstract. The source code of the proposed method will be released in a month or so on GitHub, where we will help researchers to reproduce the results in this paper.
>
> # $\mathbb{R}$ instead of $R$
> Thanks for the suggestion. We will revise this in the equations.
>
> # Summary
>
> We will make necessary revisions to the paper according to the above discussions, and prepare the public source code as soon as possible.

---

### Official Review · Reviewer_MpiZ · 2021-07-18

**Rating:** 6
**Confidence:** 5

**Summary:**

This paper applies Transformer to the matching problem and metric learning. The direct applications of ViT and the vanilla Transformer, and two solutions adapted specifically for matching images through attention. A new simplified decoder is proposed for efficient image matching, focusing on similarity computation and mapping.

**Limitations And Societal Impact:**

The authors discussed the social concerns of person re-id.



**Main Review:**


Originality: Middle

(+) This paper is to compare and extend the prevalent ViT[6] and DeTR[2], which contain both self-attention and cross (gallery-probe)-attention, in the person re-id.

(-) There are many works that use self-attention for person re-id. Such existing works are not mentioned at all. For example,

[A] Z.Zhang et al., Relation-Aware Global Attention for Person Re-identification, CVPR2020.
[B] R.Hou et al., Interaction-and-Aggregation Network for Person Re-identification, CVP2019.

(-) There are several cross-attention works on person re-id; approach which maximizes the similarity of local features of gallery and probe pair. Such existing works are not explained. For example,

[C] S.Zhao et al., Do Not Disturb Me: Person Re-identification Under the Interference of Other Pedestrians, ECCV2020


Quality: Middle

(+) Direct application of Transformer and naïve application of Transformer to image matching problem are compared.

(+) The accuracies are better, and the training time is shorter than the compared naïve application of Transformer.

(+) The proposed method shows better rank-1 rates and mAP than the results reported on state-of-the-art domain generalized person re-id works.

(-)  Details of baseline architecture are not clear.
- ViT is an image classification model. Is the same binary cross-entropy loss in GS sampler applied for all methods? Is the same MLP head applied as the proposed model?
- For Transformer-Cross, what is the value $V$ in the decoder in Eq.(1) ? Why are the Resnet query features directly used (line155) without encoders, unlike the proposed model?

(-) The performance of Transformer-Cat in Table 2 is lower than vanilla Transformer. How about appending the results of the Transformer in the lower block?

(-) The proposed method applies many improvements to the Transformer. For example,
- The proposed method uses the shared FC parameter to query and key (line-174-175).
- Multi-scale similarity scores are summed in the proposed model as in Eq.(8).

However, Table 2 compares only with the final model. Thus, we can not see how much each component improved the naive application of the Transformer. We can see the ablation study, but improvement to the naïve application of Transformer is more understandable by adding the new component to Transformer-Cross.

(+) This paper shows that replacing the first encoder directly with the ResNet feature map improves the results while being more efficient (line 241-242 and A.2).

(-) Why is only the first encoder removed? Does the replaced place  and the number of replaced encoders affect performance?


Clarity: Middle

(+)  The paper is easy to understand.

(-)  What loss function is used should be made clear. GS sampler seems to be the negative sampling method and not loss function.

(-)  It would be better to clarify MLPHead_1 and MLPHead_2 in Figure1.

Line 289-290, “We start with Dot Product + GMP + MLPHead_1”. MLPHead_1 seems to be MLPHead_2.

When MLPHead_2 is not used, the output dimension seems to be D. How can we output the score?
When MLPHead_1 is not used, the input dimension to FC3 seems to be changed to hw.


(-) For the ablation study, the author used their own measure mACC, which is averages of rank-1 rates and mAP on different datasets and runs. It is not clear if the average of different measures is possible. Also, we cannot compare the results of Table2 and Table3 because it is not used in Table2.

--
Significance: Middle

(+) The Transformer is a timely topic. ViT[6] and DeTR[2] are prevalent in computer vision.

(+) This paper clearly outperforms the state-of-the-art performance on domain generalized person re-id.

(-) The theoretical contribution is limited. This paper has made only engineering to improve the rank-1 rates and mAP and computational time.



**Time Spent Reviewing:**

10hours

---

> ### Author Response · Authors · 2021-08-10
> **Author Feedback**
>
> Dear MpiZ,
>
>
>
> Thanks for your time and effort, and thanks for the valuable comments. We shall address your concerns point by point as follows.
>
>
>
> # Related Work
>
> Indeed there are many attention based studies in person re-identification. We should have discussed them in the paper. However, they are mostly self-attention based methods, which are targeting at improving single-image's feature representation. Thanks for pointing out [C]. It is interesting to see the application of cross-attention in [C]. However, from Figs. 3&4 of [C] it can be seen that the cross-attention is still used for feature refinement, instead of distance metric learning between gallery and probe images. This paper focuses on direct image matching and metric learning between gallery and probe images, and a particular interest is to study whether Transformers and their variants are capable for this task. This is in line with the motivation of ViT and DeTR. Thanks for the suggestions. We will append a review on attention based person re-identification, including the suggested papers [A,B,C].
>
>
>
> # mAcc Measure
>
> There are three reasons that we use mAcc.
>
> * It is a unified measure, which is convenient for algorithm comparison. Both Rank-1 and mAP are accuracy measures ranging from 0%-100%, thus averaging them is possible. Besides, if a method's mAcc is 1% higher than another method, on average it means that every single measure on each dataset has been increased by 1%, which is a perceptible achievement.
> * Some algorithms perform unstably across different runs, thus the average among four runs is a more stable measure.
> * Using a unified measure is convenient, concise, and space-saving for ablation study, parameter analysis, and rebuttal.
>
> For the sake of comparison, we provide mAcc for Table 2 as follows.
>
> * ViT: 27.42%
> * Transformer: 27.01%
> * TransMatcher: 44.29%
> * Transformer-Cat: 25.34%
> * Transformer-Cross: 36.70%
> * TransMatcher-Small: 42.12%
>
>
>
> # Details of Baseline Architecture
>
> ## ViT
>
> * Yes, the same binary cross-entropy loss in GS sampler is applied for all methods.
> * ViT only contains Transformer encoders, and the same vanilla Transformer encoders are used for both ViT and the proposed method. The MLP head for the proposed model is only used in the decoder. Besides, the structure of the MLP head is almost the same as the feed-forward layer in the ViT encoder, where two FCs are used, together with ReLU and normalization.
>
> ## Transformer-Cross
>
> * As in vanilla Transformers, both K and V are the same and they are both from the encoded memory (gallery features).
>
> * Using query features directly is because
>
>   * Almost all components in Transformer encoders are already designed in decoders, such as the self-attention module (prior to cross-attention), the feed-forward layer, etc.
>   * Besides, this is also to be consistent to existing Transformers (e.g. vanilla Transformer and DeTR). We did not see a method inputting query tokens to encoders before decoders.
>   * For the proposed method, first, we think query and gallery are exchangeable and so we would like  to design a symmetric distance metric. Second, the decoder is simplified, without a self-attention module.
>
>   We did not consider that this would be the key to Transformer-Cross, but you are right, let's see what's the difference. By applying the encoders to query features prior to decoders, the mAcc is 33.77%, which is lower than 36.70% in Table 2. This may be because this structure is too complex to learn an effective metric.
>
>
>
> # Fusion in Transformer-Cat
>
> As per the concern, we tried appending the results of the Transformer in the lower block. At first, we tried score fusion as in the proposed method, resulting in an mAcc of 24.92%, compared to 25.34% in Table 2. Then we thought maybe the final normalization layer in the default encoder hindered the improvement; after removing it we got 25.10%. Later, we tried feature fusion instead of score fusion, then we got 25.22%. In any case, we were not able to improve the results by multi-layer fusion. This may be because the Transformer-Cat structure itself is not suitable for metric learning, as explained in Lines 161-166 in the paper.
>
>
>
> # Improved Components on Transformer-Cross
>
> As in ViT and DeTR, we would like to see what's the capability of the original Transformers for distance metric learning. Therefore, we did as little modifications as possible to the baseline architectures. Besides, both Transformer-Cat and Transformer-Cross are heavy to compute, which limits their values and further developments. But you are right, let's see what's the maximal capability of Transformer-Cross.
>
> * For multi-scale fusion, the mAcc is 29.52%, lower than the 36.70% in Table 2, and we observe that the learning is not stable across different runs. Again, this may be because the structure of Transformer-Cross does not directly target on image matching and metric learning.
> * For shared FC, this is the most interesting finding we learn from your feedback. Thanks. Previously, we set shared FC in the proposed model just because the distance metric is required to be symmetric. We did not observe and did not expect it to be critical in the proposed pipeline. On the other hand, Transformer-Cross is already not a symmetric design for gallery-query pairs, therefore, two different FCs are reasonable, as in the vanilla Transformer. However, now when we force the FC to be shared on both Q and K, surprisingly, the mAcc becomes 41.56%, which is much better than 36.70% in Table 2, and it is only slightly worse than the 42.12% of the proposed method. We guess this is because forcing shared FC makes the feature space of Q and K being consistent before cross-attention, and therefore it helps the subsequent metric learning task. Nevertheless, compared to the proposed model, the Transformer-Cross is still too heavy, costing too much memory and computation time.
> * Based on the shared FC, we tried the multi-scale fusion again, and got 41.52% for mAcc.
>
>
>
> # Replacement of Encoders
>
> This may be confusing. We do not remove any encoder layer or modify the default encoder structure. Instead, the actual meaning is that, if three decoder layers are used, normally there should be three corresponding encoder layers, as shown in Fig. 1. Then, to save some computation, we only use two encoder layers, together with the CNN feature map as layer0 to meet the requirement of the decoders. So the actual difference is whether we use a default encoder with the layer-number parameter being 3 or 2. Since stacked encoders work sequentially, we don't think arbitrarily removing a certain layer is necessary and reasonable.
>
>
>
> # Clarity
>
> ## Loss Function
>
> Thanks. It is the same binary cross entropy loss used in the QAConv-GS [9].
>
> ## MLP Head
>
> Thanks for the suggestions. You are right, in Line 289 it should be "Dot Product + GMP + MLPHead_2", and the input dimension to FC3 needs to be adapted to hw accordingly.
>
>
>
> # Engineering Paper
>
> No, we completely do not agree. This paper does have theoretical contributions. We point out through theoretical analysis that the original attention mechanism in Transformers is primarily designed for global feature aggregation, which is not naturally suitable for image matching and metric learning (Sections 3&4, and especially the first paragraph of Section 5). This is also validated by experiments. We believe that the analyses and discussions in this paper will be of particular interests to researchers who are concerning Transformers' capability for image matching and metric learning.
>
>
>
> # Summary
>
> Hope the above have addressed your concerns. We will make necessary revisions to the paper according to the above discussions. In summary, this paper provides a new view, from the point of image matching and metric learning requiring pairs of images (which is quite different from other popular tasks, such as image classification and object detection which only involve a single image), on how to understand Transformers, what's its capability and limitations on such task, and how to efficiently adapt it, with faster speed and less memory consumption. We hope the above weaknesses have been addressed and the rating can be re-evaluated. Thanks.

---

### Decision · Program_Chairs · 2021-09-27

**Decision:**

Accept (Poster)

**Comment:**

This paper copes with the interesting problem of using transformer-based models for image matching. This work shows that such a mechanism can successfully be used for person re-identification. The reviewers have recognised the simplicity of the proposed approach and good simple baselines described along with the approach. The experimental results show improvement over recent work on domain generalized person re-id. Despite the lack of important theoretical novelty, the tackled topic is timely and the proposed model simple. I suggest the acceptance of this work.